# Hypergraph Learning-Based Semi-Supervised Multi-View Spectral Clustering

Geng Yang [1], Qin Li [1,*], Yu Yun [1,2], Yu Lei [1,2] and Jane You [3]

1. School of Software Engineering, Shenzhen Institute of Information Technology, Shenzhen 518172, China; yangg@sziit.edu.cn (G.Y.); yuyun@stu.xidian.edu.cn (Y.Y.); 22011110198@stu.xidian.edu.cn (Y.L.)
2. School of Telecommunications Engineering, Xidian University, Xi'an 710071, China
3. Department of Computing, The Hong Kong Polytechnic University, Hong Kong 100872, China; csyjia@comp.polyu.edu.hk
* Correspondence: liqin@sziit.edu.cn

**Abstract:** Graph-based semi-supervised multi-view clustering has demonstrated promising performance and gained significant attention due to its capability to handle sample spaces with arbitrary shapes. Nevertheless, the ordinary graph employed by most existing semi-supervised multi-view clustering methods only captures the pairwise relationships between samples, and cannot fully explore the higher-order information and complex structure among multiple sample points. Additionally, most existing methods do not make full use of the complementary information and spatial structure contained in multi-view data, which is crucial to clustering results. We propose a novel hypergraph learning-based semi-supervised multi-view spectral clustering approach to overcome these limitations. Specifically, the proposed method fully considers the relationship between multiple sample points and utilizes hypergraph-induced hyper-Laplacian matrices to preserve the high-order geometrical structure in data. Based on the principle of complementarity and consistency between views, this method simultaneously learns indicator matrices of all views and harnesses the tensor Schatten p-norm to extract both complementary information and low-rank spatial structure within these views. Furthermore, we introduce an auto-weighted strategy to address the discrepancy between singular values, enhancing the robustness and stability of the algorithm. Detailed experimental results on various datasets demonstrate that our approach surpasses existing state-of-the-art semi-supervised multi-view clustering methods.

**Keywords:** semi-supervised learning; multi-view clustering; hypergraph learning

## 1. Introduction

Clustering is a fundamental task in machine learning and pattern recognition [1] that aims to partition given samples into several meaningful groups. With the rapid advancement of computer technology and the proliferation of various digital devices, a vast amount of multi-view data has emerged. Multi-view data generally refers to diverse information distinguished by attributes, sources, and characteristics of objects. Clustering methods that utilize multi-view data [2,3] can take advantage of the consistent and complementary information present in the data, resulting in learning outcomes that are more effective and accurate compared to using a single view of the data.

Multi-view clustering has become a major topic in artificial intelligence and pattern recognition over the past two decades, with numerous related methods being developed. For instance, Yang and Hussain [4] extended unsupervised k-means (UKM) clustering to a multi-view k-means clustering, called unsupervised multi-view k-means (U-MV-KM). The proposed U-MV-KM algorithm can automatically find an optimal number of clusters without any initialization for clustering multi-view data. Xia et al. [5] presented a robust multi-view spectral clustering scheme (RMSC) for mining valid information within graphs. This method considers potential noise in the input data and employs low-rank and sparse

decomposition to determine the shared transfer probability matrix. After that, a standard Markov chain model is applied to the learned transfer probability matrix for clustering. In the work of Nie et al. [6], an auto-weighted multiple graph learning (AMGL) method was proposed which automatically and without human intervention assigns reasonable weights to each graph. Peng et al. [7] developed a cross-view matching clustering (COMIC) method for multi-view clustering that automatically learns almost all parameters. This approach eliminates heterogeneous gaps between multiple views and learns view-specific representations, thereby improving the clustering process. Furthermore, it has been found that algorithms can achieve impressive performance by incorporating deep metric learning networks into multi-view clustering [8,9]. Considering the scalability and generalization of the spectral embedding, Shaham et al. introduced a deep learning approach to spectral clustering and proposed a network called SpectralNet [10], which learns a map that embeds input data points into the eigenspace of their associated graph Laplacian matrix and subsequently clusters them.

Although the abovementioned studies have achieved satisfactory results, they have all been unsupervised. In reality, complex data structures, data noise, or data corruption can significantly impact the performance of unsupervised clustering methods. Moreover, due to different user preferences, there may be more than one reasonable clustering result for the same dataset. Consequently, it is crucial to incorporate a small amount of label information into multi-view clustering.

However, labeled samples are often scarce in practical applications, and only a limited amount of label information can be used to address clustering problems. As a result, semi-supervised clustering methods have emerged. Semi-supervised clustering aims to incorporate limited prior information into clustering algorithms to satisfy user preferences and enhance data partitioning accuracy. Through the persistent efforts of researchers, semi-supervised multi-view clustering methods [11,12] have made significant progress in both theoretical research and practical applications, with a continuous stream of novel related algorithms being proposed. For instance, Liang et al. [13] introduced a method called graph-regularized partially shared non-negative matrix factorization (GPSNMF), which preserves the inherent geometric structure of the data while leveraging the available label information. Bai et al. [14] improved the label propagation algorithm to utilize label information and pairwise constraints simultaneously. Their proposed method addresses the misalignment problem of label propagation, manages pairwise constraints, and effectively explores and mines various types of prior knowledge.

Although existing semi-supervised multi-view clustering methods have achieved satisfactory results, there are several limitations that need to be addressed:

(1) Certain methods, such as AMGL [6], depend heavily on predefined graphs of different views. Clustering performance can be significantly compromised when the quality of the graphs is poor. Moreover, the ordinary graphs used by most methods only consider pairwise relationships, inevitably leading to information loss when dealing with practical problems involving complex data structures.

(2) Each view in the multi-view data contains specific attribute information, and different views exhibit consistency and complementarity. Only by comprehensively utilizing these diverse data can the essence of the subjects be fully reflected. However, certain clustering algorithms assume that all views have identical indicator matrices, and may not fully consider the differences among different views. This lack of consideration can lead to underfitting in practical applications.

(3) Several existing methods fail to fully exploit the spatial structure and complementary information contained in the indicator matrices of multiple views, even though this information is essential for improving the accuracy of multi-view clustering.

We present an efficient hypergraph learning-based semi-supervised multi-view spectral clustering approach to address the aforementioned challenges. A hypergraph is a generalization of a graph in which an edge can connect more than two vertices. Compared with ordinary graphs, hypergraphs can offer more accurate representation of the relation-

ships between objects exhibiting multiple associations. Our method combines hypergraph learning and semi-supervised multi-view spectral clustering into a cohesive framework that effectively preserves complementary information and higher-order structures by constructing hypergraph-induced hyper-Laplacian matrices on view-specific graphs and applying the tensor Schatten $p$-norm to the tensor formed from the indicator matrix. We outline our key contributions as follows:

- Our proposed method adaptively learns the graph for each view to avoid overdependence of clustering performance on predefined graphs. Furthermore, this approach considers the relationships between multiple sample points in the graph to prevent the loss of valuable information, preserving higher-order geometric structures through hypergraph-induced hyper-Laplacian matrices.
- The proposed method concurrently learns the indicator matrices of all views. It employs the tensor Schatten $p$-norm to extract these views' complementary information and low-rank spatial structures. As a result, the learned common indicator matrix offers an effective representation of the clustering structure.
- In our proposed method, we design a straightforward auto-weighted scheme for the tensor Schatten $p$-norm which adaptively determines the ideal weighted vector to handle differences between singular values. This enhances the flexibility and stability of the algorithm in practical applications. Comprehensive experiments on a wide range of datasets demonstrate the superiority of our proposed approach.

## 2. Related Works

Clustering methods are mainly divided into subspace-based methods, matrix decomposition based methods, and graph-based methods, as shown in Figure 1. As one of the most extensively researched multi-view clustering techniques, graph-based clustering has exceptionally characterized sample relationships and elucidated intricate data structures. At the heart of graph-based clustering [15,16] lies the construction of high-quality graphs, which has been the focus of numerous studies.

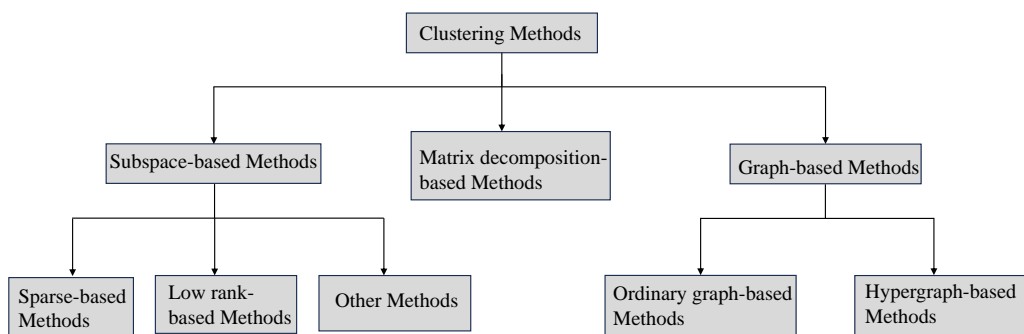

**Figure 1.** The taxonomy of clustering methods.

Currently, numerous graph-oriented multi-view clustering methods have been proposed. Recognizing that different views have consistent clustering structures, Kumar et al. [17] developed a co-regularized multi-view spectral clustering method. The indicator matrices of all views were jointly optimized to achieve consistent clustering results by applying co-regularization while following the performance of standard spectral clustering for each view. Considering the impact of similarity matrix construction and weighting strategy design on multi-view clustering, Cheng et al. [18] fused multiple graphs to obtain a more meaningful similarity matrix and leveraged known information to set corresponding weights for each view. Their method fully accounts for the differences between each view, enhancing the algorithm's stability and providing a supplementary direction for various graph-based clustering methods. Cai et al. [19] developed a multi-modal spectral clustering algorithm (MMSC) which combines diverse features to generate a common graph

Laplacian matrix. In addition, they introduced non-negative relaxation in the objective function to directly obtain the discrete cluster indicator matrix and ensure the algorithm's convergence. To properly integrate multiple data representations to achieve the optimal linear combination, Karasuyama et al. [20] proposed a sparse multiple graph integration algorithm (SMGI). This method effectively avoids the impact of noisy graphs on clustering reliability by computing sparsity weights. To improve algorithmic flexibility, Cai et al. [21] presented an adaptive multi-modality semi-supervised clustering algorithm (AMMSS) that automatically assigns appropriate weights to different modalities and learns a common indicator matrix representing the class of unlabeled samples. To enhance clustering performance through reliable graphs, Zhan et al. [22] introduced a multi-view clustering algorithm for graph learning (MVGL). This algorithm reflects clustering results by ensuring that the global graph integrated from each view has an explicit number of connected components through a rank constraint on the Laplacian matrix. Nie et al. [23] proposed a multi-view learning method with an adaptive neighbor (MLAN), which can automatically assign the optimal weight to each view without requiring additional parameters or human intervention. This method directly divides the learned ideal graph into specific clusters through reasonable rank constraints, improving the algorithm's effectiveness. To achieve a breakthrough in efficiency, Zhang et al. developed a fast multi-view semi-supervised learning method (FMSSL) [24]. By combining anchor-based graphs with multi-view semi-supervised learning strategies, FMSSL reduces the computational complexity of clustering and enhances algorithm effectiveness by exploiting feature and label information.

As is evident from the literature, graph-oriented methods have been extensively investigated and demonstrate promising performance thanks to their ability to uncover hidden structures within data. However, the aforementioned clustering approaches are all based on ordinary graphs. Many real-world problems involve highly complex data structures, and ordinary graphs, which focus solely on relationships between two sample points, often fail to provide sufficient information. Hypergraphs possess powerful characterization ability, allowing them to effectively model various networks, systems, and data structures with intricate interlocking relationships. Consequently, several methods attempt to incorporate hypergraph learning into clustering research to overcome the limitations of ordinary graphs.

Aiming to explore relationships among multiple sample points, Zhou et al. [25] extended spectral clustering techniques typically employed on ordinary graphs to hypergraphs. They further designed transductive inference and hypergraph embedding methods based on the hypergraph Laplacian. Gao et al. [26] integrated sparse coding and hypergraph learning within a unified framework, taking full advantage of hypergraphs' higher-order structure and significantly enhancing the robustness of the coding results. Yin et al. [27] incorporated hypergraph Laplacian regularization into low-rank representation to obtain a global low-dimensional representation matrix while accounting for the nonlinear geometric structure of the data. Xie et al. [28] proposed a hyper-Laplacian regularized multilinear multi-view to improve the clustering performance of a multi-view nonlinear feature subspace self-representation algorithm (HLR-M$^2$VS). This algorithm employs tensor low-rank and hyper-Laplacian regularization to preserve the global consensus and local high-order geometrical structure.

## 3. Notation and Background

### 3.1. Notation

This section lays out the notation and definitions utilized throughout the paper. Bold calligraphy, bold uppercase, bold lowercase, and lowercase letters are used to indicate tensors, matrices, vectors, and elements, respectively. For example, $\mathcal{F} \in \mathbb{R}^{n_1 \times n_2 \times n_3}$ denotes three-order tensors, $\mathbf{F} \in \mathbb{R}^{n_1 \times n_2}$ is a matrix, $\mathbf{f} \in \mathbb{R}^{n_1}$ is an $n_1$-dimensional vector, and $f_{ijk}$ represents the entries of $\mathcal{F}$. Moreover, $\mathbf{F}^{(i)}$ denotes the $i$-th frontal slice of $\mathcal{F}$. We obtain $\overline{\mathcal{F}}$ by taking the discrete Fast Fourier transform (FFT) of the tensor $\mathcal{F}$ along the third dimension, that is, $\overline{\mathcal{F}} = fft(\mathcal{F}, [], 3)$. Obviously, the inverse FFT of $\overline{\mathcal{F}}$ along the third

dimension can be represented as $\mathcal{F} = ifft(\overline{\mathcal{F}}, [], 3)$. The Frobenius norm of tensor $\mathcal{F}$ is specified by $\|\mathcal{F}\|_F = \sqrt{\sum_{i,j,k} |f_{ijk}|^2}$. The trace of matrix $\mathbf{F}$ is represented by $tr(\mathbf{F})$, and $\mathbf{I}$ is an identity matrix.

**Definition 1** (t-product [29]). *Suppose that $\mathcal{X} \in \mathbb{R}^{n_1 \times m \times n_3}$ and $\mathcal{Y} \in \mathbb{R}^{m \times n_2 \times n_3}$; then, the t-product $\mathcal{X} * \mathcal{Y} \in \mathbb{R}^{n_1 \times n_2 \times n_3}$ is defined as*

$$\mathcal{X} * \mathcal{Y} = ifft(\text{bdiag}(\overline{XY}), [], 3), \tag{1}$$

*where we use $\overline{X} = \text{bdiag}(\overline{\mathcal{X}})$ to denote the block diagonal matrix with blocks that are frontal slices of $\overline{\mathcal{X}}$.*

By using the t-product, we have the following new product decompositions of tensors (to save space, the definitions of the orthogonal tensor, f-diagonal tensor, and tensor transpose are omitted, though see [29]).

**Definition 2** (t-SVD [29]). *The tensor Singular Value Decomposition (t-SVD) of $\mathcal{F} \in \mathbb{R}^{n_1 \times n_2 \times n_3}$ is provided by $\mathcal{F} = \mathcal{U} * \mathcal{S} * \mathcal{V}^{\mathrm{T}}$, where $\mathcal{U}$ and $\mathcal{V}$ are orthogonal tensors of size $n_1 \times n_1 \times n_3$ and $n_2 \times n_2 \times n_3$, respectively, $\mathcal{S}$ is an f-diagonal tensor of size $n_1 \times n_2 \times n_3$, and $*$ denotes the t-product.*

### 3.2. Hypergraph Preliminaries

Graphs can be used to illustrate the pairwise relationship between research objects. Nevertheless, real-world problems are often described by extremely elaborate relationships between data. In [25,28], it has been shown that compressing the elaborate relationships between data into simple pairwise relationships undoubtedly results in lost information that could be useful for clustering tasks. One way to remedy the problem of information loss occurring in ordinary graphs is to represent the data in hypergraph form. It is worth noting that a hypergraph can consider the relationship between multiple vertices, allowing the high-order information and complex relationship of the data to be explored. For a given hypergraph $\mathbf{Z} = (\mathbf{V}, \mathbf{E}, \mathbf{W})$, $\mathbf{V} = \{v_i\}_{i=1}^n$ represents the set of $n$ vertices in the hypergraph, and $\mathbf{E} = \{e_j\}_{j=1}^t$ denotes the set of hyperedges, each of edge $e_j$ of which can be connected to any number of vertices. Values can be assigned to each hyperedge to build a weighted hypergraph based on a specific problem. In general, hyperedge $e_j$ is assigned a non-negative number $w(e_j)$, which is the $j$-th diagonal element of the weight matrix $\mathbf{W} \in \mathbb{R}^{t \times t}$. The incidence matrix $\mathbf{H} \in \mathbb{R}^{n \times t}$ clearly and concisely represents the interrelationship between hyperedges and vertices, and is made up of the following entries:

$$h(v_i, e_j) = \begin{cases} 1, & if \ v_i \in e_j \\ 0, & otherwise \end{cases} \tag{2}$$

Figure 2 displays a hypergraph structure comprised of 3 hyperedges and 8 vertices. As can be seen from Figure 2a, hyperedge $e_1$ connects a set of vertices $\{v_2, v_4, v_8\}$, hyperedge $e_2$ connects vertices $\{v_3, v_5, v_7\}$, and hyperedge $e_3$ connects three vertices $\{v_1, v_6, v_7\}$. The incidence matrix $\mathbf{H}$ in Figure 2b corresponds to the hypergraph of Figure 2a, which succinctly represents the connectivity of vertices and hyperedges.

Following the above definitions, we can calculate the unnormalized hyper-Laplacian matrix [25] as follows:

$$\mathbf{L}_h = \mathbf{D_V} - \mathbf{HWD_E^{-1}H^T} \tag{3}$$

where $\mathbf{D_E}$ and $\mathbf{D_V}$ indicate the degree matrices, the diagonal elements of which respectively correspond to the degree of each hyperedge $e_j$ and the degree of each vertex $v_i$. Based on the discussion presented above, it is evident that an edge in a hypergraph can connect to more than two vertices. An ordinary graph is a specialized form of a hypergraph in which each

edge contains only two vertices. Unlike ordinary graphs, hypergraphs offer more accurate representation of the relationships between objects exhibiting multiple associations.

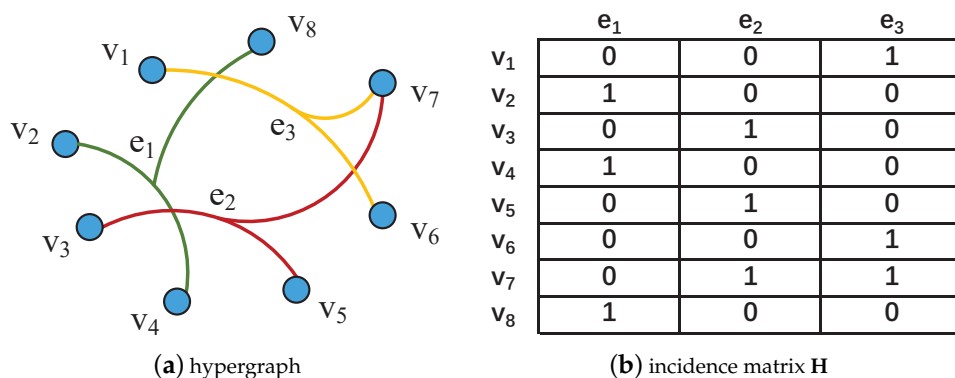

| | $e_1$ | $e_2$ | $e_3$ |
|---|---|---|---|
| $v_1$ | 0 | 0 | 1 |
| $v_2$ | 1 | 0 | 0 |
| $v_3$ | 0 | 1 | 0 |
| $v_4$ | 1 | 0 | 0 |
| $v_5$ | 0 | 1 | 0 |
| $v_6$ | 0 | 0 | 1 |
| $v_7$ | 0 | 1 | 1 |
| $v_8$ | 1 | 0 | 0 |

(**a**) hypergraph      (**b**) incidence matrix **H**

**Figure 2.** Example of (**a**) the structure of a hypergraph and (**b**) its corresponding incidence matrix **H**.

## 4. Methodology

### 4.1. Auto-Weighted Multiple Graph Learning (AMGL)

Before presenting our method, it is worth reviewing the AMGL [6], which is one of the most influential semi-supervised multi-view clustering methods that facilitate multiple graph learning. Let $\mathbf{X}^{(v)} \in \mathbb{R}^{d_v \times n}(v = 1, 2, \ldots m)$ represent the multi-view data matrix with $m$ views, where $d_v$ denotes the dimensionality of the $v$-th view and $n$ indicates the number of samples in dataset. Here, $\mathbf{X}^{(v)}$ contains $l$ tagged data and $u$ untagged data that can be divided into $c$ categories. Suppose that $\mathbf{G}^{(v)} = \left\{ g_{ij}^{(v)} \right\} \in \mathbb{R}^{n \times n}$ is the adjacency matrix of the $v$-th view and the corresponding Laplacian matrix is $\mathbf{L}_{\mathbf{G}^{(v)}} = \mathbf{D}_{\mathbf{G}^{(v)}} - ((\mathbf{G}^{(v)})^T + \mathbf{G}^{(v)})/2$, where $\mathbf{D}_{\mathbf{G}^{(v)}} \in \mathbb{R}^{n \times n}$ represents the degree matrix with $\sum_j (g_{ij}^{(v)} + g_{ji}^{(v)})/2$ as the $i$-th diagonal element. Assume that $\mathbf{F} = [\mathbf{f}_1; \mathbf{f}_2; \ldots; \mathbf{f}_n] \in \mathbb{R}^{n \times c}$ is the cluster indicator matrix, where $c$ denotes the number of classes. Thus, the objective function of AMGL can be represented by

$$\min_{\mathbf{F} \in \mathbb{R}^{n \times c}} \sum_{v=1}^{m} \sqrt{tr(\mathbf{F}^T \mathbf{L}_{\mathbf{G}^{(v)}} \mathbf{F})} \qquad (4)$$
$$\text{s.t.} \quad \mathbf{f}_i = \mathbf{y}_i \quad \forall i = 1, 2, \ldots, l$$

The label matrix $\mathbf{Y}_l = [\mathbf{y}_1; \mathbf{y}_2; \ldots; \mathbf{y}_l]$, where $\mathbf{y}_i \in \mathbb{R}^{1 \times c}$ denotes the label vector of the $i$-th data point. If the $i$-th sample falls under the $j$-th class, its element is represented by $y_{ij} = 1$; otherwise, it is represented by $y_{ij} = 0$.

Although AMGL achieves promising results, it has a number of deficiencies: (1) it cannot model unpaired relationships effectively in complex scenes, and its performance heavily relies on predefined graphs, limiting its practical applications. (2) AMGL requires the same indicator matrix $\mathbf{F}$ for different views, which is too strict and may lead to underfitting. This constraint can reduce clustering accuracy, especially for datasets with pronounced differences between views. (3) AMGL ignores the spatial structure and complementary information in the indicator matrices for multiple views, which is imperative when seeking to strengthen clustering capability.

### 4.2. Problem Formulation and Objective Determination

In this paper, we propose a semi-supervised multi-view clustering model that addresses the aforementioned problems. To be more precise, the proposed method is inspired by the tensor Schatten $p$-norm (TSP) [30,31] and combines the insights mentioned above. It simultaneously learns the indicator matrix $\mathbf{F}^{(v)} \in \mathbb{R}^{n \times c}$ of all views and leverages the TSP regularizer on the tensor $\mathcal{F} \in \mathbb{R}^{n \times m \times c}$ to encode the main spatial structure and complementary information contained in multiple views. The three-mode tensor $\mathcal{F}$ is constructed

by merging a different matrix $\mathbf{F}^{(v)}$ and then rotating its dimensionality to $n \times m \times c$ (see Figure 3). For the constructed tensor $\mathcal{F}$, its $m$-th frontal slice $\mathbf{\Delta}^{(m)}$ describes the similarity between $n$ sample points and the $m$-th cluster in different views. The idea indicator matrix $\mathbf{F}^{(v)}$ should ensure that the relationship between the $n$ data points and the $m$-th cluster is consistent in different views. Because different views usually show different cluster structures, we impose a tensor Schatten $p$-norm minimization constraint on tensor $\mathcal{F}$, ensuring that each $\mathbf{\Delta}^{(m)}$ has low-rank spatial structure. Thus, $\mathbf{\Delta}^{(m)}$ well encodes discriminative information and utilizes the complementary information embedded in inter-views, which helps to improve the clustering results. For all data points, we need to randomly select $l$ points as the labeled samples, while the remaining $u$ points correspond to the unlabeled samples. For greater convenience of derivation and description, we rearrange all the data points such that the first $l$ points are labeled samples and the subsequent $u$ points correspond to unlabeled samples. This processing ensures the randomness of data point selection and while not affecting the result. Thus, $\mathbf{F}^{(v)}$ is split into two parts, $\mathbf{F}_l^{(v)}$ and $\mathbf{F}_u^{(v)}$, i.e., $\mathbf{F}^{(v)} = \left[ \mathbf{F}_l^{(v)}; \mathbf{F}_u^{(v)} \right]$, $\mathbf{F}_l^{(v)} \in \mathbb{R}^{l \times c}$, $\mathbf{F}_u^{(v)} \in \mathbb{R}^{u \times c}$, and $l + u = n$. On this basis, we rewrite (4) as

$$\min_{\mathbf{F}_l^{(v)} = \mathbf{Y}_l} \alpha \sum_{v=1}^{m} \sqrt{tr\left(\left(\mathbf{F}^{(v)}\right)^{\mathrm{T}} \mathbf{L}_{\mathbf{G}^{(v)}} \mathbf{F}^{(v)}\right)} + \|\mathcal{F}\|_{\omega,\circledS p}^p \tag{5}$$

where $\alpha$ is the balance parameter and $\|\mathcal{F}\|_{\omega,\circledS p}$ denotes the weighted tensor Schatten $p$-norm [32] of the tensor $\mathcal{F}$, which can be defined as

$$\|\mathcal{F}\|_{\omega,\circledS p} = \left( \sum_{i=1}^{c} \left\| \overline{\mathbf{F}}^{(i)} \right\|_{\omega,\circledS p}^p \right)^{\frac{1}{p}} = \left( \sum_{i=1}^{c} \sum_{j=1}^{\min(n,m)} \omega_j * \sigma_j\left(\overline{\mathbf{F}}^{(i)}\right)^p \right)^{\frac{1}{p}} \tag{6}$$

where $\sigma_j(\overline{\mathbf{F}}^{(i)})$ is the $j$-th largest singular value of $\overline{\mathbf{F}}^{(i)}$ and $\omega_j$ is the $j$-th element of the weighted vector $\omega$. We can take advantage of the power processing scheme to bring the rank of the learned consensus indicator matrix as close to the target rank as possible by considering $0 < p \le 1$ as the power parameter.

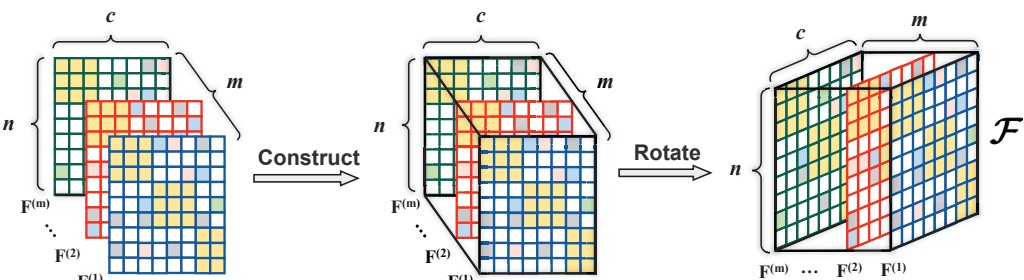

**Figure 3.** Tensor $\mathcal{F}$ construction.

In (5), it is necessary to consider the difference between singular values to fully explore the higher-order tensorial structure and set a reasonable weighted vector $\omega$. However, due to the complexity and unpredictability of the data distribution, predefining the weighted vector $\omega$ manually is not easy or circumscribed. Therefore, an adaptive weighting strategy is introduced to solve this problem. The relevant lemma is as follows.

**Lemma 1.** *For minimization of the weighted tensor Schatten $p$-norm, a closed-form global minimizer can be acquired if the singular values are in non-increasing order and the weighted values are in non-decreasing order [32,33].*

There are significant differences between the singular values of a tensor. Because large singular values can usually describe the main structure of the data, these values should

shrink less. Based on the above insights and Lemma 1, an effective automatic weighting strategy is designed to improve the algorithm's flexibility. Specifically, the *j*-th element of the weighted vector $\omega$ is defined as

$$\omega_j = \frac{\sqrt{n \times m}}{\sigma_j + \psi}, \tag{7}$$

where $\sqrt{n \times m}$ is an empirical value, *n* and *m* respectively represent the number of samples and the number of views, $\sigma_j$ denotes the *j*-th singular value, and $\psi$ is a very small value which is set to $10^{-6}$ in the experiment. It is worth noting that the weighted value $\omega_j$ indicates the degree to which the *j*-th singular value shrinks, that is, the larger singular value should shrink less and the smaller singular value should shrink more. As can be seen from Figure 4, if the singular values are in non-increasing order, the corresponding weighted values $\omega_j$ are in non-decreasing order. In other words, this strategy can automatically calculate a reasonably weighted vector $\omega$ for different datasets, improving the flexibility and stability of the algorithm.

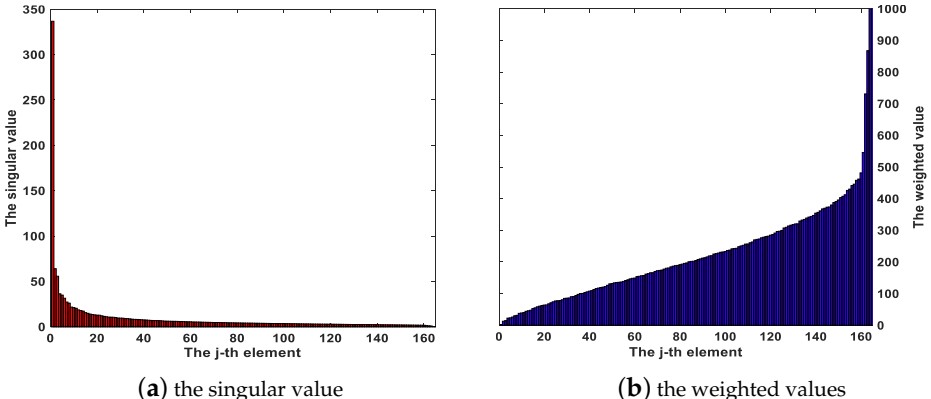

(**a**) the singular value      (**b**) the weighted values

**Figure 4.** Illustration of the singular values and weighted values.

In terms of (5), the Lagrangian function can be expressed as follows:

$$\alpha \sum_{v=1}^{m} \sqrt{tr\left(\left(\mathbf{F}^{(v)}\right)^{\mathrm{T}} \mathbf{L}_{\mathbf{G}^{(v)}} \mathbf{F}^{(v)}\right)} + \|\mathcal{F}\|_{\omega,\circledS\!p}^{p} + \zeta(\mathrm{Y}, \mathcal{F}) \tag{8}$$

where the formalized term $\zeta(\mathrm{Y}, \mathcal{F})$ derives from the constraints and Y represents the Lagrange multipliers. Setting the derivative concerning $\mathbf{F}^{(v)}$ to zero provides us with

$$\alpha \sum_{v=1}^{m} \lambda^{(v)} \frac{\partial tr\left(\left(\mathbf{F}^{(v)}\right)^{\mathrm{T}} \mathbf{L}_{\mathbf{G}^{(v)}} \mathbf{F}^{(v)}\right)}{\partial \mathbf{F}^{(v)}} + \frac{\partial \|\mathcal{F}\|_{\omega,\circledS\!p}^{p}}{\partial \mathbf{F}^{(v)}} + \frac{\partial \zeta(\mathrm{Y}, \mathcal{F})}{\partial \mathbf{F}^{(v)}} = 0, \tag{9}$$

where

$$\lambda^{(v)} = \frac{1}{2\sqrt{tr\left(\left(\mathbf{F}^{(v)}\right)^{\mathrm{T}} \mathbf{L}_{\mathbf{G}^{(v)}} \mathbf{F}^{(v)}\right)}} \tag{10}$$

It is worth noting that in (10) there is a dependency of $\lambda^{(v)}$ on $\mathbf{F}^{(v)}$; therefore, (9) is inconvenient to solve directly. Inspired by previous multi-view clustering approaches [6,34], if $\lambda^{(v)}$ is set to be stationary, then (9) can be regarded as the solution to (11) provided below:

$$\min_{\mathbf{F}_l^{(v)} = \mathbf{Y}_l} \alpha \sum_{v=1}^{m} \lambda^{(v)} tr\left(\left(\mathbf{F}^{(v)}\right)^{\mathrm{T}} \mathbf{L}_{\mathbf{G}^{(v)}} \mathbf{F}^{(v)}\right) + \|\mathcal{F}\|_{\omega,\circledS\!p}^{p} \tag{11}$$

For (11), it is challenging to manually obtain a suitable graph to represent the correlation between samples due to the complexity of the data distribution and the differences between the views. Moreover, if the predefined graph is not adequate, the performance of the clustering algorithm may become severely degraded. In order to overcome this limitation, we introduce adaptive learning graphs to our model, which can enhance its stability by avoiding dependency on a fixed graph.

Furthermore, we adopt the hypergraph learning approach and use hypergraph-induced hyper-Laplacian matrices in our method to reveal the higher-order geometric structure of the data. This strategy can effectively capture the relationship between multiple sample points, preserve essential information from the data, and improve the stability and flexibility of the clustering method. Therefore, we formulate our objective function as follows:

$$\min_{\mathbf{F}_l^{(v)} = \mathbf{Y}_l} \sum_{v=1}^{m} \frac{1}{\tau^{(v)}} \|\mathbf{G}^{(v)} - \mathbf{S}^{(v)}\|_F^2 + \beta \|\mathcal{F}\|_{\omega, \mathbb{S}_p}^p$$
$$+ \alpha \sum_{v=1}^{m} \lambda^{(v)} tr\left((\mathbf{F}^{(v)})^{\mathrm{T}} \mathbf{L}_h^{(v)} \mathbf{F}^{(v)}\right) \tag{12}$$
$$\text{s.t.} \quad \mathbf{S}^{(v)} \geq 0, \ \mathbf{S}^{(v)}\mathbf{1} = \mathbf{1}, \ \tau^{(v)} \geq 0, \ \sum_{v=1}^{m} \tau^{(v)} = 1$$

where $\mathbf{S}^{(v)} \in \mathbb{R}^{n \times n}$ is the learned graph for the $v$-th view, which represents the affinities among all the samples. For the sake of avoiding the negative effects of scale, we impose the non-negative normalization constraint $0 \leq \mathbf{S}^{(v)} \leq 1$, $\mathbf{S}^{(v)}\mathbf{1} = \mathbf{1}$, to maintain a unified range of values for the affinities of all sample points via regression. Here, $\beta$ represents a trade-off parameter and $\tau^{(v)}$ describes the magnitude of the non-negative normalized weight for the $v$-th view.

In Formula (12), $\mathbf{L}_h^{(v)}$ represents the hyper-Laplacian matrix of the $v$-th view constructed from the affinity matrix $\mathbf{S}^{(v)}$. Because $\mathbf{S}^{(v)}$ is adaptively learned, hypergraphs constructed based on the affinity matrix $\mathbf{S}^{(v)}$ do not depend on fixed predefined graphs; in this way, the relationship between data points can be better described. When $\mathbf{L}_h^{(v)}$ in the model is replaced by $\mathbf{L}_{\mathbf{S}^{(v)}}$, it becomes an ordinary Laplacian matrix constructed based on $\mathbf{S}^{(v)}$, and the corresponding hyper-Laplacian regularization is simplified into the traditional graph Laplacian constraint that can only capture the pair relationship between samples.

To be more specific, in the process of building the hyper-Laplacian matrix $\mathbf{L}_h^{(v)}$ we regard each sample in the multi-view dataset as a vertex in the hypergraph, take each vertex $p_i^{(v)}$ as the centroid vertex, and construct the corresponding hyperedge $e_j^{(v)}$ using the k-nearest neighbor method; that is to say, the hypergraph constructed for each view is composed of $n$ hyperedges and each hyperedge contains $k$ vertices. Accordingly, the view-specific incidence matrix $\mathbf{H}^{(v)} \in \mathbb{R}^{n \times n}$ can be obtained according to Formula (2), that is, in the $v$-th view, we have $h_{ij}^{(v)} = 1$ if vertex $p_i^{(v)}$ belongs to hyperedge $e_j^{(v)}$ and $h_{ij}^{(v)} = 0$ otherwise. Because the affinity matrix $\mathbf{S}^{(v)}$ of each view is adaptively learned in Formula (12), a weight $w(e_j^{(v)})$ can be assigned to each hyperedge $e_j^{(v)}$ on each view according to the following formula:

$$w(e_j^{(v)}) = \sum_{p_i^{(v)} \in e_j^{(v)}} S_{ij}^{(v)} \tag{13}$$

The corresponding weight matrix $\mathbf{W}^{(v)} \in \mathbb{R}^{n \times n}$ is a diagonal matrix with $w(e_j^{(v)})$ as its diagonal element, i.e., $\mathbf{W}^{(v)} = \text{diag}\left(w(e_1^{(v)}), w(e_2^{(v)}), \ldots, w(e_n^{(v)})\right)$. Based on this, the edge

degree $\mathbf{D}_{\mathbf{E}}^{(v)}$ and the vertex degree $\mathbf{D}_{\mathbf{P}}^{(v)}$ corresponding to the $v$-th view can be calculated according to the following formula:

$$
\begin{aligned}
d(e_j^{(v)}) &= \sum_{i=1}^{n} h(p_i^{(v)}, e_j^{(v)}) \\
d(p_i^{(v)}) &= \sum_{j=1}^{n} w(e_j^{(v)}) h(p_i^{(v)}, e_j^{(v)})
\end{aligned}
\tag{14}
$$

where $\mathbf{D}_{\mathbf{E}}^{(v)}$ and $\mathbf{D}_{\mathbf{P}}^{(v)}$ are diagonal matrices with diagonal elements that correspond to the degrees of each hyperedge $e_j^{(v)}$ and the degrees of each vertex $p_i^{(v)}$, respectively.

Finally, the view-specific hyper-Laplacian matrix $\mathbf{L}_h^{(v)}$ built on the affinity matrix $\mathbf{S}^{(v)}$ can be formulated as follows:

$$
\mathbf{L}_h^{(v)} = \mathbf{D}_{\mathbf{P}}^{(v)} - \mathbf{H}^{(v)} \mathbf{W}^{(v)} \mathbf{D}_{\mathbf{E}}^{(v)^{-1}} \mathbf{H}^{(v)^{\mathrm{T}}}
\tag{15}
$$

To sum up, each vertex in the hypergraph constructed in this paper is connected with at least one hyperedge, and each hyperedge is associated with a weight. The proposed model first adaptively learns the affinity matrix $\mathbf{S}^{(v)}$ of each view, then fully considers the relationship between multiple sample points, constructs the hypergraph using the acquired affinity matrix, and preserves the higher-order geometric structure through the hypergraph-induced hyper-Laplacian matrix $\mathbf{L}_h^{(v)}$. In this way, the model realizes the effective exploration of the higher-order information and complex structure in the data while avoiding dependence on predefined graphs.

*4.3. Optimization*

It is infeasible to solve (12) directly due to the presence of multiple variables. The Augmented Lagrange Multiplier (ALM) method is an efficient solver for the above model (12). In order to solve optimization problems with separable structures, the Alternating Direction Method of Multipliers (ADMM) is proposed, which is used to obtain the solution of the global problem by coordinating the solutions of the subproblems. To adopt an alternating direction minimization strategy with the model, we first introduce auxiliary variables $\mathcal{J}$ and $\mathbf{K}^{(v)}$ to replace $\mathcal{F}$ and $\mathbf{S}^{(v)}$, respectively, that is to say, $\mathcal{F} = \mathcal{J}$ and $\mathbf{S}^{(v)} = \mathbf{K}^{(v)}$. Then, we substitute $\mathcal{F}$ and $\mathbf{S}^{(v)}$ into (12); by simple algebraic operations, we can then rewrite (12) as the following augmented Lagrangian function:

$$
\begin{aligned}
\min_{\mathbf{F}_l^{(v)} = \mathbf{Y}_l} \; & \alpha \sum_{v=1}^{m} \lambda^{(v)} tr\left( (\mathbf{F}^{(v)})^{\mathrm{T}} \mathbf{L}_h^{(v)} \mathbf{F}^{(v)} \right) + \beta \| \mathcal{J} \|_{\omega, \circledS}^{p} \\
& + \frac{\mu}{2} \left\| \mathcal{F} - \mathcal{J} + \frac{\mathcal{Q}}{\mu} \right\|_F^2 + \sum_{v=1}^{m} \frac{1}{\tau^{(v)}} \| \mathbf{G}^{(v)} - \mathbf{K}^{(v)} \|_F^2 \\
& + \sum_{v=1}^{m} \frac{\gamma}{2} \left\| \mathbf{S}^{(v)} - \mathbf{K}^{(v)} + \frac{\mathbf{T}^{(v)}}{\gamma} \right\|_F^2 \\
\text{s.t.} \quad & \mathbf{S}^{(v)} \geq 0, \; \mathbf{S}^{(v)} \mathbf{1} = \mathbf{1}, \; \tau^{(v)} \geq 0, \; \sum_{v=1}^{m} \tau^{(v)} = 1
\end{aligned}
\tag{16}
$$

where $\mathcal{Q}$ and $\mathbf{T}^{(v)}$ are used to indicate Lagrange multiplier while $\mu > 0$ and $\gamma > 0$ denote the penalty factor. We use an alternate optimization strategy to solve (16), which involves the following subproblems.

- **Solving $\mathbf{K}^{(v)}$ with fixed $\mathbf{S}^{(v)}$, $\mathbf{T}^{(v)}$, and $\frac{1}{\tau^{(v)}}$.** Now, the optimization problem with respect to $\mathbf{K}^{(v)}$ in (16) can be simplified as

$$
\min_{\mathbf{K}^{(v)}} \sum_{v=1}^{m} \frac{1}{\tau^{(v)}} \| \mathbf{G}^{(v)} - \mathbf{K}^{(v)} \|_F^2 + \sum_{v=1}^{m} \frac{\gamma}{2} \left\| \mathbf{S}^{(v)} - \mathbf{K}^{(v)} + \frac{\mathbf{T}^{(v)}}{\gamma} \right\|_F^2
\tag{17}
$$



Because $\{\mathbf{K}^{(v)}\}_{v=1}^{m}$ are independent, we can solve each $\mathbf{K}^{(v)}$ independently. Setting the derivative to zero with respect to $\mathbf{K}^{(v)}$, we obtain

$$-\frac{2}{\tau^{(v)}}\mathbf{G}^{(v)} + (\gamma\mathbf{I} + \frac{2}{\tau^{(v)}}\mathbf{I})\,\mathbf{K}^{(v)} - \gamma(\mathbf{S}^{(v)} + \frac{\mathbf{T}^{(v)}}{\gamma}) = 0 \tag{18}$$

By simple algebra, the optimal solution for $\mathbf{K}^{(v)}$ is provided by

$$\mathbf{K}^{(v)} = \frac{\gamma\mathbf{S}^{(v)} + \mathbf{T}^{(v)} + \frac{2}{\tau^{(v)}}\mathbf{G}^{(v)}}{\gamma + \frac{2}{\tau^{(v)}}} \tag{19}$$

- **Solving $\mathbf{S}^{(v)}$ with fixed $\mathbf{K}^{(v)}$ and $\mathbf{T}^{(v)}$.** According to (16), the solution to this subproblem can be calculated as follows:

$$\min_{\mathbf{S}^{(v)}} \sum_{v=1}^{m} \frac{\gamma}{2}\|\mathbf{S}^{(v)} - \frac{\gamma\mathbf{K}^{(v)}-\mathbf{T}^{(v)}}{\gamma}\|_F^2 \tag{20}$$
$$\text{s.t.} \quad \mathbf{S}^{(v)} \geq 0, \ \mathbf{S}^{(v)}\mathbf{1} = \mathbf{1}$$

It should be noted that there is no dependence between all $\mathbf{S}^{(v)}(v = 1,\dots,m)$ in (20). The closed-form solution $\mathbf{S}^{(v)^{*}}$ of each $\mathbf{S}^{(v)}$ is provided by $\mathbf{S}^{(v)^{*}}(i,:) = (\frac{\mathbf{B}^{(v)}(i,:)}{\gamma}+\varrho\mathbf{1})_+$ [35], where $\mathbf{B}^{(v)} = \gamma\mathbf{K}^{(v)} - \mathbf{T}^{(v)}$ and $\varrho$ is the Lagrangian multiplier.

- **Solving $\mathbf{F}^{(v)}$ with fixed $\mathcal{J}$, $\mathcal{Q}$, $\mathbf{L}_{\mathrm{h}}^{(v)}$, and $\lambda^{(v)}$.** At this point, the optimization problem in (16) with respect to $\mathbf{F}^{(v)}$ can be formulated as

$$\min_{\mathbf{F}_l^{(v)}=\mathbf{Y}_l} \alpha \sum_{v=1}^{m} \lambda^{(v)} tr((\mathbf{F}^{(v)})^{\mathrm{T}}\mathbf{L}_{\mathrm{h}}^{(v)}\mathbf{F}^{(v)}) + \frac{\mu}{2}\|\mathcal{F} - \mathcal{J} + \frac{\mathcal{Q}}{\mu}\|_F^2 \tag{21}$$

We denote $\mathbf{F}^{(v)} = \begin{bmatrix} \mathbf{Y}_l; \mathbf{F}_u^{(v)} \end{bmatrix}$, $\mathbf{J}^{(v)} = \begin{bmatrix} \mathbf{J}_l^{(v)}; \mathbf{J}_u^{(v)} \end{bmatrix}$, $\mathbf{Q}^{(v)} = \begin{bmatrix} \mathbf{Q}_l^{(v)}; \mathbf{Q}_u^{(v)} \end{bmatrix}$, and $\mathbf{L}_{\mathrm{h}}^{(v)} = \begin{bmatrix} \mathbf{L}_{ll}^{(v)} & \mathbf{L}_{lu}^{(v)} \\ \mathbf{L}_{ul}^{(v)} & \mathbf{L}_{uu}^{(v)} \end{bmatrix}$, then substitute them into (21). Moreover, the fact that $\{\mathbf{F}^{(v)}\}_{v=1}^{m}$ are independent allows each $\mathbf{F}^{(v)}$ to be solved independently. Therefore, by simple algebra, (21) becomes

$$\begin{aligned} \min_{\mathbf{F}^{(v)}} \alpha\lambda^{(v)}tr(&\begin{bmatrix} \mathbf{Y}_l \\ \mathbf{F}_u^{(v)} \end{bmatrix}^{\mathrm{T}} \begin{bmatrix} \mathbf{L}_{ll}^{(v)} & \mathbf{L}_{lu}^{(v)} \\ \mathbf{L}_{ul}^{(v)} & \mathbf{L}_{uu}^{(v)} \end{bmatrix} \begin{bmatrix} \mathbf{Y}_l \\ \mathbf{F}_u^{(v)} \end{bmatrix}) \\ +\frac{\mu}{2}&\left\|\begin{bmatrix} \mathbf{Y}_l \\ \mathbf{F}_u^{(v)} \end{bmatrix} - \begin{bmatrix} \mathbf{J}_l^{(v)} \\ \mathbf{J}_u^{(v)} \end{bmatrix} + \frac{1}{\mu}\begin{bmatrix} \mathbf{Q}_l^{(v)} \\ \mathbf{Q}_u^{(v)} \end{bmatrix}\right\|_F^2 \\ = \text{Const} + 2\alpha\lambda^{(v)}&tr((\mathbf{F}_u^{(v)})^{\mathrm{T}}\mathbf{L}_{ul}^{(v)}\mathbf{Y}_l) + \alpha\lambda^{(v)}tr((\mathbf{F}_u^{(v)})^{\mathrm{T}}\mathbf{L}_{uu}^{(v)}\mathbf{F}_u^{(v)}) \\ +\frac{\mu}{2}tr((\mathbf{F}_u^{(v)})&^{\mathrm{T}}\mathbf{F}_u^{(v)}) - \mu tr((\mathbf{F}_u^{(v)})^{\mathrm{T}}(\mathbf{J}_u^{(v)} - \frac{\mathbf{Q}_u^{(v)}}{\mu})) \end{aligned} \tag{22}$$

Setting the derivative to zero with respect to $\mathbf{F}_u^{(v)}$, we obtain the $v$-th class indicator $\mathbf{F}_u^{(v)}$ for the unlabeled data as follows:

$$\mathbf{F}_u^{(v)^{*}} = (\alpha\lambda^{(v)}\mathbf{L}_{uu}^{(v)} + \frac{\mu}{2}\mathbf{I})^{-1}(\frac{\mu}{2}(\mathbf{J}_u^{(v)} - \frac{\mathbf{Q}_u^{(v)}}{\mu}) - \alpha\lambda^{(v)}\mathbf{L}_{ul}^{(v)}\mathbf{Y}_l) \tag{23}$$

- **Solving $\mathcal{J}$ with fixed $\mathcal{F}$ and $\mathcal{Q}$.** In this case, the solution to this subproblem can be simplified as follows:

$$\min_{\mathcal{J}} \ \beta\|\mathcal{J}\|^p_{\omega,\circledS\!\mathcal{P}} + \frac{\mu}{2}\left\|\mathcal{F} - \mathcal{J} + \frac{\mathcal{Q}}{\mu}\right\|^2_F \tag{24}$$

To solve (24), we need the following theorem.

**Theorem 1** ([32]). *For $\mathcal{A} \in \mathbb{R}^{n_1 \times n_2 \times n_3}$, $h = \min(n_1, n_2)$, which has t-SVD $\mathcal{A} = \mathcal{U} * \mathcal{S} * \mathcal{V}^T$, the optimal solution of (25) is*

$$\arg\min_{\mathcal{X}} \ \frac{1}{2}\|\mathcal{X} - \mathcal{A}\|^2_F + \varepsilon\|\mathcal{X}\|^p_{\omega,\circledS\!\mathcal{P}} \tag{25}$$

*which can be written as*

$$\mathcal{X}^* = \Gamma_{\varepsilon*\omega}(\mathcal{A}) = \mathcal{U} * ifft(\mathbf{P}_{\varepsilon*\omega}(\bar{\mathcal{A}})) * \mathcal{V}^T \tag{26}$$

*where $\mathbf{P}_{\varepsilon*\omega}(\overline{\mathcal{A}}) \in \mathbb{R}^{h \times h \times n_3}$ is an f-diagonal tensor with diagonal elements that can be found using the GST algorithm introduced in Lemma 1 of [32] and $\overline{\mathcal{A}} = ifft(\mathcal{A}, [\ ], 3)$.*

According to the above two theorems, the solution of (24) is

$$\mathcal{J}^* = \Gamma_{\frac{\beta}{\mu}*\omega}\left(\mathcal{F} + \frac{\mathcal{Q}}{\mu}\right). \tag{27}$$

- **Solving $\tau^{(v)}$ with other fixed variables.** According to (16), this subproblem can be solved by

$$\min_{\tau^{(v)}} \sum_{v=1}^{m} \frac{\|\mathbf{G}^{(v)} - \mathbf{S}^{(v)}\|^2_F}{\tau^{(v)}}, \ \text{s.t.} \ \sum_{v=1}^{m} \tau^{(v)} = 1, \tau^{(v)} \geq 0 \tag{28}$$

In the case of $e^{(v)} = \|\mathbf{G}^{(v)} - \mathbf{S}^{(v)}\|_F$, (28) can be expressed as

$$\min_{\tau^{(v)}} \sum_{v=1}^{m} \frac{e^{(v)^2}}{\tau^{(v)}}, \quad \text{s.t.} \ \sum_{v=1}^{m} \tau^{(v)} = 1, \tau^{(v)} \geq 0 \tag{29}$$

Considering that $\sum_{v=1}^{m} \tau^{(v)} = 1$, based on the Cauchy-Schwartz inequality, we can find the following:

$$\sum_{v=1}^{m} \frac{e^{(v)^2}}{\tau^{(v)}} = \left(\sum_{v=1}^{m} \frac{e^{(v)^2}}{\tau^{(v)}}\right)\left(\sum_{v=1}^{m} \tau^{(v)}\right) \geqslant \left(\sum_{v=1}^{m} e^{(v)}\right)^2 \tag{30}$$

Equation (30) only holds if $\sqrt{\tau^{(v)}} \propto \frac{e^{(v)}}{\sqrt{\tau^{(v)}}}$. Because the right-hand side of (30) is the constant, the optimal solution for $\tau^{(v)}$ ($v = 1, 2, \ldots, m$) can be deduced as follows:

$$\tau^{(v)} = e^{(v)} / \sum_{i=1}^{m} e^{(i)} \tag{31}$$

- **Solving $\lambda^{(v)}$ with fixed $\mathbf{F}^{(v)}$.** Similar to $\lambda^{(v)}$ in (11), $\lambda^{(v)}$ in (16) can be updated by

$$\lambda^{(v)} = \frac{1}{2\sqrt{\text{tr}\left(\left(\mathbf{F}^{(v)}\right)^T \mathbf{L}_h^{(v)} \mathbf{F}^{(v)}\right)}} \tag{32}$$

- **Update $\mathbf{T}^{(v)}$, $\mathcal{Q}$, $\gamma$, and $\mu$.** Below are the formulas for updating these variables:

$$\mathbf{T}^{(v)} = \mathbf{T}^{(v)} + \gamma(\mathbf{S}^{(v)} - \mathbf{K}^{(v)}) \tag{33}$$

$$\mathcal{Q} = \mathcal{Q} + \mu(\mathcal{F} - \mathcal{J}) \tag{34}$$

$$\gamma = \min(\rho\gamma, \gamma_{\max}) \tag{35}$$

$$\mu = \min(\rho\mu, \mu_{\max}) \tag{36}$$

where $\mu_{\max}$, $\gamma_{\max}$, and $\rho$ are constants.

After obtaining $\mathbf{F}_u^{(v)}$, we can obtain discrete labels for unlabeled data using

$$l_i = \arg\max_{j \in \{1,2,\dots,c\}} \mathbf{F}_u(i,j), \quad \forall i = 1, 2, \dots, u \tag{37}$$

where $\mathbf{F}_u(i,j)$ refers to the $i$-th row and $j$-th column item of $\mathbf{F}_u = (\sum_{v=1}^{m} \lambda^{(v)} \mathbf{F}_u^{(v)})/m$.

We denote the discrete label matrix $\mathbf{Y}_u = [\mathbf{y}_1; \mathbf{y}_2; \dots; \mathbf{y}_u]$, with elements $y_{ij} = 1$ if $l_i = j$ ($j = 1, 2, \dots c$) and $y_{ij} = 0$ otherwise.

Presented in Algorithm 1 is the complete pseudocode needed to solve (12).

---

**Algorithm 1** Hypergraph Learning-Based Semi-Supervised Multiview Spectral Clustering

---

**Input**: Graph $\mathbf{G}^{(v)}$ for $m$ views, label matrix $\mathbf{Y}_l$, the cluster number $c$, parameter $\alpha$, $\beta$ and $p$.
**Output**: Label information for unlabeled samples.

1: Initialize $\mathbf{K}^{(v)} = \mathbf{T}^{(v)} = 0$, $\mathbf{S}^{(v)} = \mathbf{G}^{(v)}$, $\lambda^{(v)} = \frac{1}{m}$, $\tau^{(v)} = \frac{1}{m}$ ($v = 1, 2, \dots, m$). $\mathcal{Q} = \mathcal{J} = \mathbf{0}$, $\mu = \gamma = 0.1$, $\rho = 2$, $\mu_{\max} = \gamma_{\max} = 10^{10}$.
2: **while** not converge **do**
3:     Construct hyper-Laplacian matrices $\{\mathbf{L}_h^{(v)}\}_{v=1}^{m}$ based on $\{\mathbf{S}^{(v)}\}_{v=1}^{m}$ by (15);
4:     Update $\{\mathbf{K}^{(v)}\}_{v=1}^{m}$ by using (19);
5:     Update $\{\mathbf{S}^{(v)}\}_{v=1}^{m}$ by solving (20);
6:     Update $\{\mathbf{F}_u^{(v)}\}_{v=1}^{m}$ by using (23);
7:     Update $\mathcal{J}$ according to (27);
8:     Update $\tau^{(v)}$ by solving (31);
9:     Update $\lambda^{(v)}$ by calculating (32);
10:    Update $\{\mathbf{T}^{(v)}\}_{v=1}^{m}$ and $\mathcal{Q}$ according to (33) and (34), respectively;
11:    Update $\gamma$ and $\mu$ according to (35) and (36), respectively;
12: **end while**
13: **Return** the indicator matrix $\mathbf{F}$.
14: Calculate label matrix $\mathbf{Y}_u$ by (37).

---

## 5. Experiment

### 5.1. Experimental Setting

Dataset. A comprehensive set of experiments was conducted to verify the validity and stability of the proposed model on four different datasets, including the following.

Yale dataset (http://vision.ucsd.edu/content/yale-face-database (accessed on 15 May 2023)). This dataset, created by Yale University, contains 165 face images of fifteen individuals. Each individual is represented by eleven grayscale images with varying postures, facial expressions, and lighting conditions. In the experiment, we chose 4096-dimensional (D) intensity features, 3304-D LBP features, and 6750-D Gabor features as three distinct types of views.

Caltech-101 dataset [36] This dataset comprises 8677 images divided into 101 classes, each containing 40 to 800 image files. For our experiment, we selected 1474 images belonging to seven categories: Snoopy, Windsor-Chair, Stop-sign, Face, Garfield, Dollar-Bill, and Motorbikes. We considered three types of features as distinct views, including 1160-D LBP features, 620-D HOG features, and 2560-D SIFT features.

ORL dataset (http://www.uk.research.att.com/facedatabase.html (accessed on 15 May 2023)). The ORL dataset includes 400 facial photographs of forty individuals, some taken at different times, resulting in variations in facial expressions, facial details, and lighting angles. We selected 6750-D Gabor features, 4096-D intensity features, and 3304-D LBP features as different views.

MSRC dataset [37] This dataset consists of 240 images in eight categories. We chose seven categories for our experiments: tree, building, cow, face, plane, car, and bicycle. We employed five visual features to construct multiple views, including 24-D CM features, 254-D Centrist features, 256-D LBP features, 512-D GIST features, and 576-D HOG features.

Comparison algorithms. In this paper, we evaluated the performance of the proposed clustering algorithm by comparing it with seven other clustering algorithms, namely, Adaptive Semi-Supervised Learning Approach for Multiple Feature Modalities (AMMSS) [21], i.e., the semi-supervised learning strategy for integration of multiple graphs based on sparse weights under label propagation (SMGI) [20], Multiple Graph Clustering Framework based on Automatic Weighting Strategy (AMGL) [6], Semi-Supervised Image Classification using Multiple-Modal Curriculum Learning (MMCL) [38], Multi-View Clustering with Adaptive Neighbors (MLAN) [23], Hyper-Laplacian Regularization-based Semi-Supervised Clustering with Multiple Views (HLR-M$^2$VS) [28], and Fast Multi-View Semi-Supervised Learning (FMSSL) [24].

Evaluation Metrics. To comprehensively evaluate clustering performance, we employed three fundamental metrics in our experiments: accuracy (ACC) [39], purity [40], and normalized mutual information (NMI) [41]. For each of these metrics, a higher value signifies better clustering capability.

Parameter Settings. Our proposed model incorporates three trade-off parameters ($\alpha$, $\beta$, and $p$) that affect the final results. To obtain optimal performance, we adjusted $\alpha$ within the range of $[0.001, 0.005, 0.01, 0.05, 0.1, 0.5, 1, 5]$, $\beta$ within the range of $[0.001, 0.005, 0.01, 0.05, 0.1, 0.5, 1]$, and the power parameter $p$ within the range of $(0, 1]$ with an interval of 0.1. For the other comparison algorithms, we followed the experimental settings described in the respective studies or adjusted the parameters for optimal results.

*5.2. Experimental Results*

Results Analysis. To discover more valid information about the unlabelled data from the limited labeled data, we randomly selected between 10% and 60% of the labeled samples to be used in semi-supervised learning. Without loss of generality, the outcome is the average value obtained after repeating each experiment 10 times. Figures 5–8 report the experimental results of the eight clustering algorithms under different label proportions on the four datasets. According to these clustering results, the following observations can be drawn.

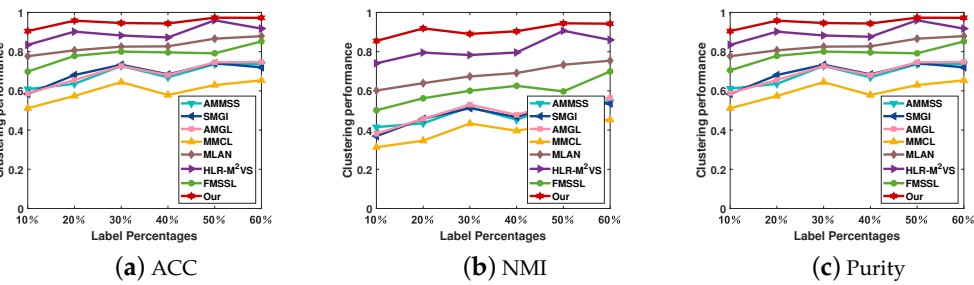

**Figure 5.** Semi-supervised clustering results on Caltech-101 dataset.

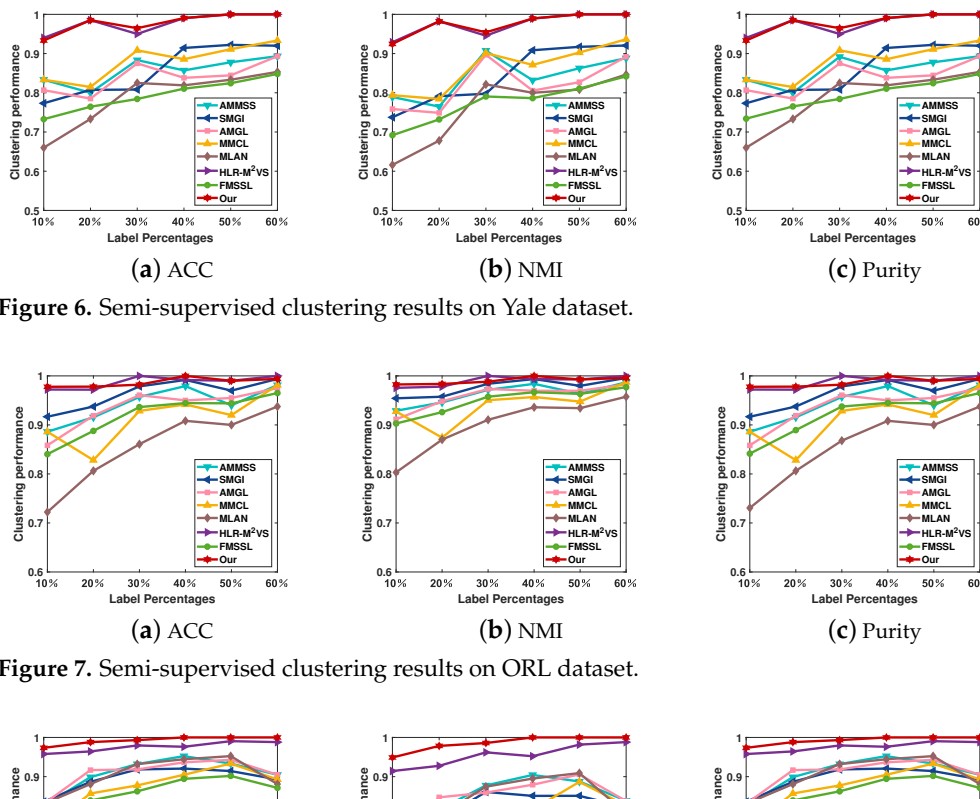

**Figure 6.** Semi-supervised clustering results on Yale dataset.

**Figure 7.** Semi-supervised clustering results on ORL dataset.

**Figure 8.** Semi-supervised clustering results on MSRC dataset.

(1) The overall performance of our algorithm and the HLR-M$^2$VS algorithm is superior to that of the other six algorithms. This is because the other algorithms only consider pairwise relationships in the graph, resulting in the loss of effective information. Our work and the HLR-M$^2$VS algorithm both focus on relationships between multiple sample points and employ hypergraph-induced hyper-Laplacian matrix to preserve higher-order geometric structures. In addition, both our work and the HLR-M$^2$VS algorithm are based on the tensor, which can unearth the complementary information and spatial structure hidden in multi-view data. The other algorithms do not consider this factor. Despite this, it is apparent from the figures that the HLR-M$^2$VS algorithm is less accurate and stable than ours. For example, on the Caltech-101 dataset the accuracy of the HLR-M$^2$VS algorithm is 7.02% lower than ours for the case containing 10% of the labeled samples. The reason for this is that the HLR-M$^2$VS algorithm assumes that all views have the same indicator matrix. It adopts the tensor nuclear norm instead of the tensor Schatten *p*-norm as the global constraint to ensure the consistency principle without considering the significant differences of all views and different singular values, resulting in poor algorithm performance in practical application.

(2) Overall, our method is significantly superior to the other seven methods on all four datasets. For example, on the Yale dataset, our method shows a remarkable increase in comparison to the MMCL of around 10.48%, 11.83%, and 10.48% in terms of ACC, NMI, and purity, respectively, for 40% labeled samples. On the MSRC dataset, for 20% labeled samples, our method shows a relative improvement of 2.38%, 5.08%, and 2.38% in terms of the ACC, NMI, and purity compared to the second-best method, HLR-M$^2$VS. Our method

emphasizes each view's role in clustering by adaptively allocating appropriate weights for different views, thereby improving the algorithm's flexibility. Moreover, our method integrates hypergraph learning and semi-supervised multi-view spectral clustering into a unified framework, and leverages the tensor Schatten $p$-norm to encode the complementary information and low-rank spatial structure. Thus, the learned indicator matrix is well able to characterize the clustering structure, and the clustering results accurately represent the categories of samples. It is worth noting that the clustering results on databases with different dimensions show that the dimension of the data affects the clustering results, and our method is sensitive to the data complexity.

(3) For the same dataset, the clustering performance of most algorithms improves with an increasing number of labeled samples. For example, as the proportion of labeled samples in the Caltech-101 dataset increases from 10% to 60%, the clustering accuracy of the AMMSS algorithm correspondingly increases from 60.90% to 73.63%, an improvement of 12.73%. Similarly, the clustering accuracy of our algorithm increases from 90.48% to 97.25%, an increase of 6.77%. This shows that semi-supervised clustering can use limited labeled data to mine hidden information in unlabeled data for better clustering performance. In addition, a large amount of prior information can enhance the ability to infer unknown labels, further improving the clustering accuracy of the algorithm.

Parameters analysis. In (12), the $\beta$ parameter is utilized to balance the proportion of the tensor Schatten $p$-norm, while the $\alpha$ parameter represents the impact of the spectral clustering term on the model. To analyze the influence of these two parameters on the clustering performance of our model, we present a 3D histogram in Figure 9 that visualizes the clustering accuracy obtained with different parameter settings on the Caltech-101, Yale, ORL, and MSRC datasets.

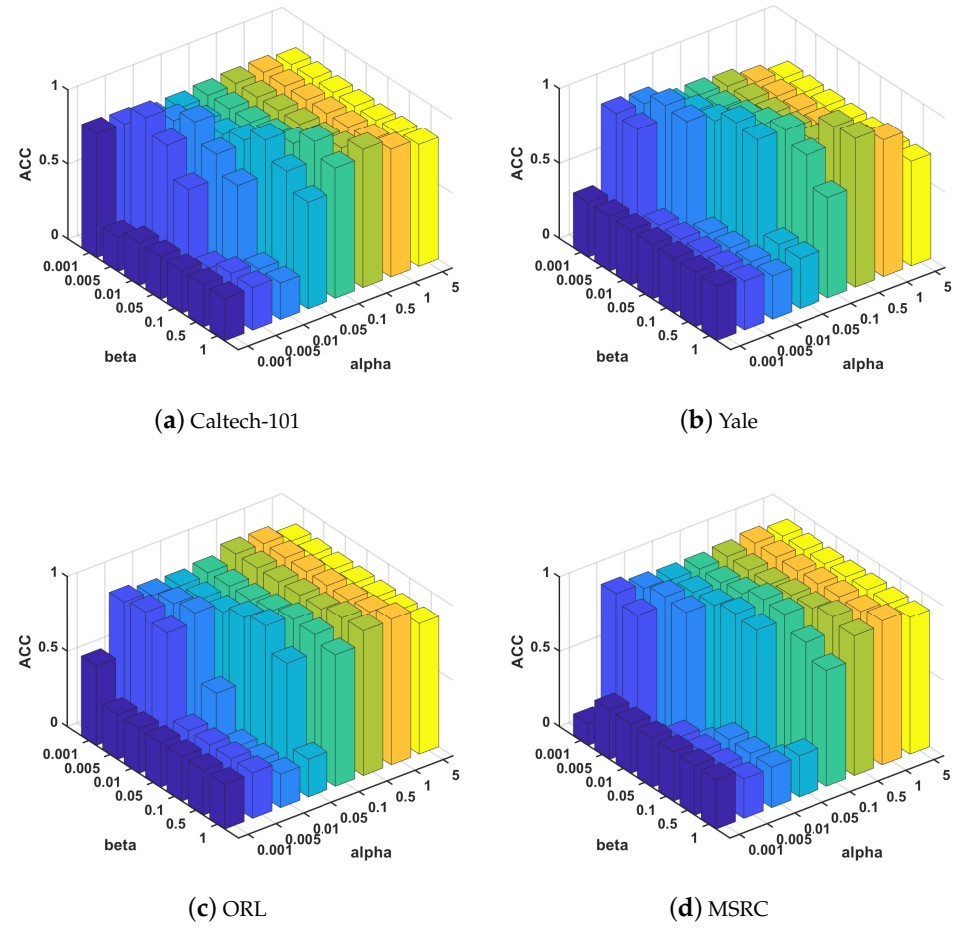

(**a**) Caltech-101

(**b**) Yale

(**c**) ORL

(**d**) MSRC

**Figure 9.** Parameter tuning ($\alpha$ and $\beta$) regarding ACC and NMI on the MSRC, ORL, and Yale datasets.

Specifically, in order to determine the most favorable results, all of our experiments were conducted with the $\alpha$ parameter in the range of [0.001, 0.005, 0.01, 0.05, 0.1, 0.5, 1, 5] and the $\beta$ parameter in the range of [0.001, 0.005, 0.01, 0.05, 0.1, 0.5, 1]. Based on the data in Figure 9, the clustering performance fluctuates significantly with varying $\alpha$ and $\beta$. Our approach achieves optimal clustering performance on the Yale dataset when $\alpha = 0.5$ and $\beta = 1$, and the case is similar for the Caltech-101, ORL, and MSRC datasets.

The experimental results are relatively stable within a specific range. When $\alpha > 0.05$, the clustering performance of our model substantially improves. This improvement may be attributed to the spectral clustering term preserving higher-order geometric structures using the hypergraph-induced hyper-Laplacian matrix, which helps the model to perform better. Furthermore, the tensor Schatten $p$-norm explores the complementary content between different views. Therefore, selecting a reasonable value for $\beta$ after determining the value of $\alpha$ contributes to higher model accuracy.

Convergence analysis. Research suggests that demonstrating the convergence of inexact ALM with three or more block variables remains an open question [42]. Consequently, it is not easy to demonstrate the convergence of Algorithm 1 theoretically. To facilitate further analysis, we recorded $\sum_{v=1}^{m} \left\| \mathbf{F}_{(t+1)}^{(v)} - \mathbf{J}_{(t+1)}^{(v)} \right\|_{\infty}$ for each iteration on the four databases presented in Figure 10. Note that $\mathbf{F}_{(t+1)}^{(v)}$ and $\mathbf{J}_{(t+1)}^{(v)}$ respectively represent the $\mathbf{F}^{(v)}$ and $\mathbf{J}^{(v)}$ matrices obtained by the $(t+1)$-th iteration. The x-axis represents the number of iterations for each sub-plot, while the y-axis corresponds to the variable error. Figure 10 illustrates that the variable error drops rapidly within relatively few iterations and stabilizes as the number of iterations increases, indicating that our model converges sufficiently.

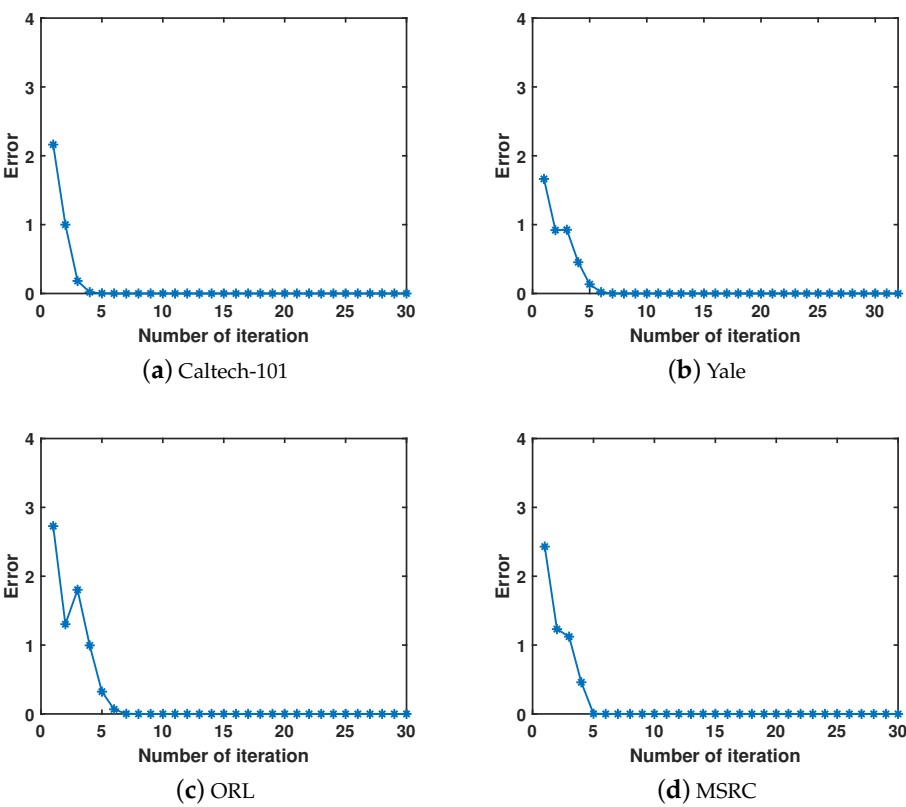

**Figure 10.** Convergence curves on the ORL and MSRC datasets.

Complexity Analysis. Our method consists of two stages, namely, construction of hypergraphs and optimization by iteratively solving Equation (16). The cost of constructing the initial k-nearest neighbors graph is $\mathcal{O}_1(mn^2 d + mn^2 log(n))$. Then, we mainly focus on the optimization of four variables, i.e., $\mathbf{F}^{(v)}$, $\mathcal{J}$, $\mathbf{S}^{(v)}$, and $\mathbf{K}^{(v)}$. For the $\mathbf{F}^{(v)}$ subproblem, it

takes $\mathcal{O}_2(mu^3 + mucn)$ in each iteration. For the $\mathcal{J}$ subproblem, calculating the $3D$ FFT and $3D$ inverse FFT of an $n \times m \times c$ tensor and $c$ SVDs of $n \times m$ matrices in the Fourier domain dominates the main computation. Because we have $n \gg m$ in the multi-view setting, the computation at each iteration takes $\mathcal{O}_3(mnclog(mn) + m^2cn)$. In terms of the $\mathbf{S}^{(v)}$ subproblem, it takes $\mathcal{O}_4(mn^2log(n))$ in each iteration. For the $\mathbf{K}^{(v)}$ subproblem, it takes $\mathcal{O}_5(mn^2)$ in each iteration. Therefore, the main computational complexity of our proposed method in each iteration is $\mathcal{O} = \mathcal{O}_1 + \mathcal{O}_2 + \mathcal{O}_3 + \mathcal{O}_4 + + \mathcal{O}_5$.

## 6. Conclusions

In this paper, we propose a hypergraph learning-based semi-supervised multi-view spectral clustering method. This method first adaptively learns the affinity matrix of each view, then fully considers the relationship between multiple sample points. It uses the learned affinity matrix to construct hypergraphs while preserving the higher-order geometric structure through the hypergraph-induced hyper-Laplacian. This technique effectively explores higher-order information and complex structures in the data while avoiding dependence on predefined graphs. Moreover, the proposed method simultaneously learns the indicator matrix for all views, and employs the tensor Schatten $p$-norm to uncover the low-rank spatial structure and complementary content hidden in these views. As a result, the learned common indicator matrix can more effectively reflect the cluster structure. We additionally design a simple auto-weighted scheme for the tensor Schatten $p$-norm which adaptively determines the ideal weighted vector to accommodate differences between singular values, thereby enhancing the algorithm's flexibility and stability in practical applications. Experiments on four real datasets demonstrate that our method outperforms cutting-edge competitors regarding overall effectiveness.

The hypergraph learning approach and automatic weighting strategy proposed in this paper can serve as valuable references for researchers in other fields. In future work, we intend to continue to investigate more effective methods for mining higher-order information and complex structures in data. In semi-supervised learning, we intend to study the ratio between labeled and total samples in order to identify the optimal proportion, enabling the algorithm to achieve the best possible clustering performance with minimal labeled data as support. The research work in this paper focuses on the construction of multiple complex relationships between data while ignoring the generalization and scalability of the algorithm. Due to its high computational complexity, the proposed algorithm is not suitable for large-scale data. In future studies, the proposed method could be improved based on the relevant theory of anchors to make it more applicable to large-scale data. On the other hand, thanks to the powerful nonlinear mapping ability and feature extraction ability of deep learning, it has become a research trend to flexibly apply deep learning to different fields. Therefore, in forthcoming research another possibility is to use the traditional clustering model with deep learning or to effectively integrate traditional methods with deep networks as a means of effectively improving clustering performance.

**Author Contributions:** Conceptualization, G.Y.; methodology, G.Y. and Q.L.; writing—original draft preparation, G.Y.; writing—review and editing, Y.Y. and Y.L.; supervision, Q.L. and J.Y.; funding acquisition, Q.L. All authors have read and agreed to the published version of the manuscript.

**Funding:** This work was supported in part by the Natural Science Foundation of Guangdong Province under Grant 2023A1515011845 and in part by the Guangdong v2x Data Security Key Technology and Expanded Application R&D Industry Education Integration Innovation Platform under Grant 2021CJP016.

**Data Availability Statement:** Publicly available datasets were analyzed in this study. These data can be found at: Yale, http://vision.ucsd.edu/content/yale-face-database; Caltech-101, https://tensorflow.google.cn/datasets/catalog/caltech101; MSRC, https://mldta.com/dataset/msrc-v1/; ORL, http://www.uk.research.att.com/facedatabase.html (accessed on 15 May 2023).

**Conflicts of Interest:** The funders had no role in the design of the study, in the collection, analysis, or interpretation of data, in the writing of the manuscript, or in the decision to publish the results.

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
