# Peer review of "Hypergraph Learning-Based Semi-Supervised Multi-View Spectral Clustering"

_electronics, doi:10.3390/electronics12194083_

Round 1
Reviewer 1 Report
1) Discuss this paper in the introduction section: Unsupervised multi-view K-means clustering algorithm.
2) Analyze the computational complexity of the proposed algorithm.
The English language would be further improved.
Reviewer 2 Report
The manuscript entitled "Hypergraph Learning Based Semi-Supervised Multi-View Spectral Clustering" presents the methodology based on Graph-based semi-supervised multi-view clustering. This approach has a clear perspective to be used for a wide range of the problems in different fields.
Strengths:
The manuscript is easy-to-follow in general, however, some places in the text evidently require additional explanation.
The results are represented clearly
The methodology is of high demand in different fields
Shortcomings:
The body of the manuscript requires the reorganization
Some important problems were not addressed at all (e.g., the generalizability and the scalability)
It is not clear how the complexity of data (dimensionality) will affect the performance
The introductory part requires revision and extending
The main comments are the following:
1. " Subsequently, standard spectral clustering is used to obtain the clustering solution. " The sentence looks as out of context
2. The figure with the taxonomy of the methods of clustering can complement the overview of the methods presented in the text.
3. The principle of a separation the overview of the literature to the part provided in the Introduction and that in the Releted works is not clear.
4. The real-world data can be characteriized with different complexity. How Authors assess the capacity of the proposed method to work with high-dimensional data and if this method is sensitive to the data complexity?
5. The problems of generalization and scalability are not addressed.
6. The methods of spectral clustering based on neural networks were not mentioned or discussed (please, see, for example, Shahom et al SpectralNet: Spectral Clustering using Deep Neural Networks. ICLR (2018)).
7. Hypergraphs: please provide the basic explanation in the introductory part. Also, the incidence matrix looks excessive (the mentioning of the vertex/edge order would be sufficient).
8. In section "Methodology" Authors introduce AMGL whilein the introductory part it was mentioned as the alternative approach. Please provide with a link to the methodology proposed by Authors. For example, you may place lines 219-221 at the beginning of the paragraph followed by adding the additional explanations.
9. " Here, F is made up of F( v)" Please, reformulate.
10. "To keep the generality intact, we rearrange all the data points so that the first l points are labeled samples, and the subsequent u points correspond to unlabeled samples." The provided explanations are not sufficient.
11. Please verify line 166 (typo?)
12. "Compressing them into simple pairwise relationships will undoubtedly result in lost information that could be useful for clustering tasks. " The sentence requires the literature-based support. In current form it looks disputable.
13. Affinity matrix S(v): please provide with the additional explanations.
14. Authors write: "It is infeasible to solve (13) directly due to the presence of multiple variables. Motivated by the augmented Lagrange multiplier (ALM) method [32], we first introduce auxiliary variables J and K( v) to replace F and S (v ) , respectively, and then substitute them into (13)." In its current formulation, the idea is not clear without the special knowledge in the field.
15. " μ > 0 and γ > 0 denote the penalty factor" The penalty or the regularizer?
16. To the abovementioned: auxiliary variables J and K( v) require the comments
Minor editing of English is recommended.
Reviewer 3 Report
This paper examines a novel approach to graph-based semi-supervised multi-view clustering, addressing limitations in existing methods. It introduces hypergraph learning and tensor Schatten p-norm to capture high-order data structures, leveraging complementarity between views. An auto-weighted strategy enhances algorithm robustness. Extensive experiments show superior performance over current methods.
Some important points have to be clarified or justified and few concerns need to be addressed by the authors for the betterment of the manuscript
1- Can you provide more insight into the motivation behind developing a hypergraph-based approach for multi-view clustering and how it addresses the limitations of existing methods?
2- It is suggested to add the following reference in the Introduction:-
- Sergio Saponara, Abdussalam Elhanashi, Alessio Gagliardi, "Reconstruct fingerprint images using deep learning and sparse autoencoder algorithms," Proc. SPIE 11736, Real-Time Image Processing and Deep Learning 2021, 1173603 (12 April 2021); https://doi.org/10.1117/12.2585707
3- Can you elaborate on the concept of "complementarity and consistency between views" and how your method leverages it to improve clustering results?
4- What is the significance of using the tensor Schatten p-norm in your approach for extracting complementary information and low-rank spatial structure from multiple views?
5- Could you provide more details about the "auto-weighted strategy"? How does it enhance the algorithm's robustness and stability?
6- How did you select the datasets for your experiments, and what criteria were used to evaluate the performance of your approach against existing state-of-the-art methods?
7- Were there any challenges or trade-offs encountered during the development of your method, and how did you address them?
8- What future research directions or extensions of your work do you envision, and how do you believe your approach could be further improved or generalized to different domains?
Further proofreading is required
Round 2
Reviewer 2 Report
After the major revision made by Authors the manuscript can be recommended to publication in Electronics (MDPI).
Reviewer 3 Report
Thanks to authors for their implementation for the manuscript